Stat-tracks and mediotypes: powerful tools for modern ichnology based on 3D models

Belvedere Matteo matteo.belvedere@gmail.com 1
Bennett Matthew R. 2
Marty Daniel 1
Budka Marcin 2
Reynolds Sally C. 2
Bakirov Rashid 2
1 Section d’Archéologie et Paléontologie, Paléontologie A16, Office de la Culture , Porrentruy , Canton Jura , Switzerland
2 Institute for Studies in Landscape and Human Evolution, Faculty of Science and Tecnology, Bournemouth University , Poole , United Kingdom
Anquetin Jérémy
Electronic publication date: 2018 Jan 11
Publication date: 2018
Volume: 6
Electronic Location ID: e4247
Received 2017 Jul 28; Accepted 2017 Dec 18
Copyright: ©2018 Belvedere et al.
Copyright year: 2018
Copyright holder: Belvedere et al.
License: This is an open access article distributed under the terms of the Creative Commons Attribution License, which permits unrestricted use, distribution, reproduction and adaptation in any medium and for any purpose provided that it is properly attributed. For attribution, the original author(s), title, publication source (PeerJ) and either DOI or URL of the article must be cited.
License URL: https://creativecommons.org/licenses/by/4.0/

Keywords: Mediotype, Stat-track, Vertebrate ichnology, Digital ichnology, Ichnotaxonomy, Digital reconstruction

Funding: Swiss Federal Roads Office PALA16 project NERC NE/H004246/1 NE/M021459/1 Alexander von Humboldt-Stiftung Fellowship This work is supported by: the Swiss Federal Roads Office (FEDRO, 95%) and the Canton Jura (5%) though PALA16 project (Matteo Belvedere, Daniel Marty), by NERC grants (NE/H004246/1 and NE/M021459/1) (Matthew R. Bennett, Marcin Budka). Matteo Belvedere was also supported during the earlier phases of this project by an Alexander von Humboldt-Stiftung Fellowship. The funders had no role in study design, data collection and analysis, decision to publish, or preparation of the manuscript.

==============================
Vertebrate tracks are subject to a wide distribution of morphological types. A single trackmaker may be associated with a range of tracks reflecting individual pedal anatomy and behavioural kinematics mediated through substrate properties which may vary both in space and time. Accordingly, the same trackmaker can leave substantially different morphotypes something which must be considered in creating ichnotaxa. In modern practice this is often captured by the collection of a series of 3D track models. We introduce two concepts to help integrate these 3D models into ichnological analysis procedures. The mediotype is based on the idea of using statistically-generated three-dimensional track models (median or mean) of the type specimens to create a composite track to support formal recognition of a ichno type. A representative track (mean and/or median) is created from a set of individual reference tracks or from multiple examples from one or more trackways. In contrast, stat-tracks refer to other digitally generated tracks which may explore variance. For example, they are useful in: understanding the preservation variability of a given track sample; identifying characteristics or unusual track features; or simply as a quantitative comparison tool. Both concepts assist in making ichnotaxonomical interpretations and we argue that they should become part of the standard procedure when instituting new ichnotaxa. As three-dimensional models start to become a standard in publications on vertebrate ichnology, the mediotype and stat-track concepts have the potential to help guiding a revolution in the study of vertebrate ichnology and ichnotaxonomy.

Introduction

This paper uses a combination of dinosaur and human tracks to explore an emerging tool in ichnology, namely the use of statistics-based virtual tracks (e.g., mean or median tracks) to explore morphological variability (i.e., departures from typical or average morphology), and its potential role in ichnotaxonomy. The reader may be forgiven for questioning at the outset, however, what dinosaur and human tracks have in common and why they appear together in the same work. Even though such tracks are imprinted by completely different trackmakers and in very different geological time periods, both are biogenic sedimentary structures that represent the dynamic interaction of a foot (morphology + kinematics) with the substrate properties at the time of formation (Padian & Olsen, 1984; Marty, Strasser & Meyer, 2009; Falkingham, 2014). Once formed they may be affected and modified during taphonomy (e.g., Cohen et al., 1991; Scott et al., 2007; Marty, Strasser & Meyer, 2009; Scott, Renaut & Owen, 2010) and diagenesis (Phillips et al., 2007). Therefore, even where a track is imprinted by a single species of trackmaker, the resulting population of tracks may be very varied and in extreme cases are classified as different morphotypes (e.g., Romilio, Tucker & Salisbury, 2013). Accordingly, in the study of ichnites it is crucial to understand and distinguish biomechanical, behavioural and preservation variants and to recognize the range of features that do (and do not) correspond to the morphological record that a given trackmaker’s foot can make in the geological record. Despite being a complex biogenic sedimentological structure, a track can often closely, although not perfectly, represent the morphology of the trackmaker’s autopodium, or portion thereof (i.e., proximal region of the autopodium may not constitute parts of the palm or sole surface) allowing the study of the geographical and/or temporal distribution of an ichnotaxon and of its trackmaker. Key features of a dinosaur track, such as the digit (phalangeal) pads and claw impressions for example may not be preserved in one single track (e.g., Razzolini et al., 2017), but different features may be preserved in several different tracks along the length of a trackway, or across a given ichnocoenosis. Understanding this variability lies at the heart of many ichnological studies, especially when identification of the trackmaker is very complicated and speculative.

Traditionally, the study of vertebrate tracks has been addressed by a combination of detailed description, photography and in some cases by blending and overlapping track outlines (e.g., the ‘composite tracks’ in Olsen & Baird, 1986). Digitization tools and procedures (e.g., optical laser scanners, close-range photogrammetry, CT scans) have increased appreciably in recent years whether generated by photogrammetry (e.g., Matthews, Noble & Breithupt, 2016, and references therein) or by laser scanning (e.g., Bates et al., 2008; Petti et al., 2008; Belvedere, Mietto & Ishigaki, 2010; Belvedere & Mietto, 2010). The capture, presentation and analysis of 3D track data has become increasingly standard in publications on vertebrate tracks and is now considered by many to be an essential part of the ichnological tool kit, although there remains a body of ‘traditionalists’ who still hold to more conventional methods. Despite these differences most would agree, however, that 3D data has a potentially revolutionary role in quantifying morphological variability. In fact, Falkingham (2016 , p. 73) has commented on the difficulty of applying objective methods to track outlines and basic descriptions and has emphasised “the importance of ichnologists taking advantage of modern digitizing techniques in order to communicate and share full three-dimensional (3-D) data”. The use of multiple 3D track models to explore morphological variation is increasingly common, but the tools to aid this remain in their infancy. There has been a number of attempts recently to develop methods for creating 3D mean or median tracks by co-registering multiple examples and computing mean depth (z) values (e.g., Crompton et al., 2012; Bennett et al., 2016a; Bennett et al., 2016b). We explore this further here via a series of case studies using one of these approaches based on the freeware DigTrace (Budka et al., 2016). To assist in this, we introduce the idea of ‘stat-track’, namely a statistical-based (e.g., mean, median, standard deviation) track created from a population of co-registered tracks that allow one to explore variability via measures of central morphological tendency and the ‘mediotype’, the mean and median stat-track generated from type specimens (e.g., holotype, paratypes) of a given ichnotaxon. It is analogous to the term ‘digitype’ introduced by Adams et al. (2010) to describe the digital facsimile of a type specimen, in the same way that ‘plastotype’ (a cast of the primary type, Morningstar, 1924) can be considered equivalent to the original specimen.

Methodology

Trackway and track terminology and labelling of the dinosaur trackways from the Ajoie ichnocoenosis follows standard approaches (Marty, 2008; Marty et al., 2010; Marty, Falkingham & Richter, 2016). Human track terminology follows Bennett & Morse (2014). The geological context and applied documentation methodology of the studied material are presented in Supplemental Information 1.

Digital methods

Digital 3D track data are obtained from a range of optical laser scanners and increasingly via digital photogrammetry (Falkingham, 2012; Bennett & Morse, 2014). Bennett et al. (2013) provide a comparative review of data derived from optical laser scanners and photogrammetry concluding that while the former gives more accurately scaled results the latter is operationally much easier.

Traditionally, tracks have been analysed by comparison of track or outlines and/or the placement and comparison of inter-landmark distances, at their simplest these may be length and width measurements. Landmark placement occurs in the field by the operator selecting measurement points. In the case of a 2D photograph, outline drawing, or 3D surface this is usually in the form of a physically located and labelled landmark. Either way basic dimensions for multiple tracks can be obtained, size distribution considered, and centres of central tendency calculated. Especially, but not only (e.g., Rodrigues & Santos, 2004; Belvedere, 2008; Castanera et al., 2013; Lallensack, van Heteren & Wings, 2016), in human ichnology, the digital placement whether on a 2D or 3D image also allows Cartesian coordinates to be recorded and subject geomorphometric analysis via a Procrustes analysis or similar approach (e.g., Berge, Penin & Pellé, 2006; Hammer & Harper, 2007; Bennett et al., 2009). Hatala et al. (2016) have advocated the use of what they call ‘areas of interest’ as an alternative approach; in truth, these are just geometrically placed landmarks across the plantar surface of a track.

The idea of ‘whole-track’ analysis, introduced by Crompton et al. (2012) in their analysis of the Laetoli hominin tracks (Tanzania), requires the registration (Goshtasby, 2005) of one or more tracks, to allow areas of anatomical similarity to be overlapped as defined by the user, or by some form of statistical parameter (e.g., least squares). The term ‘registration’ refers to the process of transforming one track to a ‘source’ (or ‘master’) track, such that the three-dimensional morphology of other tracks is optimally overlapped. Once a succession of tracks has been registered, it is possible to compare the depth along the ‘z’-axis values for each track, and thereby compute measures of central tendency for the population of registered tracks. A mean, or median, track can be created in this way and they provide, in theory at least, a more accurate topological representation of the trackmaker’s foot impression than any one individual track. Being aware of the possibility that some features can be washed out by this process, the occurrence of peculiar characteristics in the mean and median tracks provides evidence of the importance of those recurrent features. If interested only in the morphology of the foot, only the best tracks of the trackway should be considered, as those with a very different preservation (e.g., with extensive collapsing of the track walls) are influencing the resulting mean track and may bias the entire analysis. A mean track, in fact, includes all intra-track variability within in a trackway caused, for example, by behaviour, variation in gait or variation in rheological properties of the substrate along a trackway (Morse et al., 2013; Razzolini et al., 2014). It therefore draws out the recurring topological (i.e., depth variation) track morphology, which by inference in theory at least should give insight into the morphological feature of the autopodium and, by comparison, into the biomechanical signature left by the trackmaker’s mode of gait.

A method to register the plantar pressure records based on statistical parametric mapping (SPM) was developed by Pataky & Goulermas (2008), and termed by the authors pedobarographic Statistical Parametric Mapping (pSPM). Registrations were achieved by various automated algorithms using a progressive approach in which tracks are registered first to an initial track (the first in the series) and then re-registered to an initial mean track. Different registration methods were compared by Pataky, Goulermas & Crompton (2008), who observed that manual methods were found to be as accurate as the automated ones when averaged between operators. Therefore, a pSPM-based approach has started being used by researchers (e.g., Crompton et al., 2012; Morse et al., 2013; Bates et al., 2013). This approach, however, is not without its limitations, when applied to a wider range of tracks: the main issue is the need of a smooth and relatively similar topology across a range of tracks to obtain an automated registration of the tracks. In reality, fossil tracks are complex structures that can contain forms, which vary between tracks, which may interfere with automated registration. To use pSPM the researcher has to intervene on the track topology by removing such distractions through cropping a track by elevation thus to focus solely on the interpreted plantar (or palm, if on a forelimb) surface. The limitations in the manual registration tools in pSPM, the absence of a simple user interface and the availability of the pSPM source code led Budka et al. (2016) to create the freeware DigTrace (http://www.digtrace.co.uk) which caters for the registration of tracks, comparison and computation of measures of central tendency using a landmark-matching process (for details see Bennett et al., 2016a; Bennett et al., 2016b).

Mediotype and stat-tracks

Stat-track

A stat-track (statistical based tracks) is defined as the virtual specimens generated from the co-registration of 3D models of two or more tracks. Stat-tracks can be used to make morphological comparisons.

There are currently at least two published software solutions for the co-registration of tracks as outlined above (i.e., DigTrace, and the unpublished pSPM code of Pataky & Goulermas, 2008). Here we use the freeware DigTrace (Budka et al., 2016) which is based on the co-registration of tracks via user-defined anatomical or morphological landmarks, essentially the matching of similar points (however defined) on two tracks. Once two or more tracks are co-registered in the x–y plane using either rigid (i.e., dependant on the shape and size of the specimen, optimal for comparing tracks of the same trackmaker or of similar size) or affine transformations (i.e., dependant only on the shape of the specimen and not on the size, optimal for comparing tracks from different trackmakers or with very different dimensions), the software computes measures of central tendency for the depth or z-values. DigTrace allows the generation of six different stat-tracks: mean, median, maximum and minimum differences, standard deviation and point-to-point comparison, all of which can be exported as point clouds (.asc or .csv files). It is important to emphasise that DigTrace is one of several potential software solutions that achieve similar ends.

Mediotype

The mediotype is a specific case of stat-track. It is defined as the mean and/or median stat-track (as from the Latin prefix medio-) generated from the co-registration of type specimens (holotype, paratype(s) and/or other type tracks).

In general, stat-tracks allow one to make reproducible comparisons of a track sample, based on the actual three-dimensional morphology of the track and not just on bi-dimensional (interpretative outline) drawings, descriptions and/or photos. Mediotypes, mean and median stat-tracks allow the study of a morphology derived from several different tracks, whereas the use of deviation-based stat-tracks (e.g., minimum, maximum or standard deviation) allows the identification of intra-sample variance with the sampled tracks (even between different mediotypes or mean-media start-tracks). These results can be used as a basis for more detailed descriptions and comparisons with other material and ichnotaxa. Software such as DigTrace also allow the comparison between different mediotypes and makes direct comparisons of holotypes with other type material. One can also explore extramorphological variations (e.g., displacement rims, collapsing substrate, dragging of the digits, etc.) on the overall shape of the track.

It is important to remember, however, that the researcher must know the purpose of the investigation as this determines both the sample selection and the interpretation of the resulting stat-track. For example, if one is interested in instituting a new ichnotaxon, he has to underline the key features and to see if they are present in different footprints; however, only the best tracks should be used (i.e., the type specimens) and the mediotype should highlight the common features. If a sample of unselected tracks (including also the poor ones) is chosen for the same purpose, the mean and median stat-tracks will more likely blur out the key features and create a useless virtual track. Also, locomotion-oriented studies may not benefit from the use of stat-tracks as they rely more on the autopodium/substrate interaction than on the autopodium morphology. It is also worth mentioning that the quality of 3D models (e.g., resolution, presence of noise, holes etc.) can influence the generation and interpretation of stat-tracks (and mediotypes), and it is in the responsibility of the researcher to select only the models that are suitable for his research and to try to avoid very low resolution or very noisy meshes.

It is also important to remember that the stat-track outputs (i.e., the .asc and .csv files) are visualized in DigTrace as false-colour maps. In order to keep the distinction between the actual specimens and the virtual tracks the researcher should use a different colour scheme to illustrate the stat-tracks. Here we have used a ‘rainbow’ scheme to visualize the mean and median stat-tracks and the mediotypes, and a black and white scheme to illustrate the standard-deviation, min and max stat-track, however there is no rule, and the files can even be imported in a different software (e.g., CloudCompare) to create different colour schemes, as long as the researcher explain the colour scheme used.

Case Studies

Here, we present some applications of the stat-track and mediotype concepts to show its appliance to morphological analyses of similar sauropod tracks from the same ichnocoenosis but with a very different size (3.1), to the morphological study of a track population of the same trackmaker species (3.2 Laetoli), to the erection of new ichnotaxa (3.3.1 Jurabrontes curtedulensis), and to the validation through comparison of new ichnotaxa (3.3.2 Megalosauripus transjuranicus). The IDs of the tracks used to generate each stat-track and mediotype of this work are listed in Table 1.

Table 1 Labels and identification numbers of the specimens used for generating stat-tracks and mediotypes.

For each figure of this work are reported the number of tracks used to generate stat-tracks and mediotypes, together with their labels. (h) indicated the holotype of an ichnotaxon; (p) indicates a paratype of an ichnotaxon. Tracksite abbreviations: BSY, Bois de Sylleux; TCH, Tchâfoué; SCR, Sur Combe Ronde. More information in the Supplemental Information 1.

Mediotype	N. of tracks	Track IDs	
Tiny sauropod
(Fig. 1A)	10	BSY1040-S20-RP8, BSY1040-S20RP9, BSY1040-S21-LP6, BSY1040-S1-RP8, BSY1040-S2-RP6, BSY1040-S2-LP7, BSY1040-S3-LP3, BSY1040-S3-LP5, BSY1040-S25-RP1, BSY1040-S6-LP3	
Medium sauropod
(Fig. 1B)	9	BSY1040-S12-LP2, BSY1040-S12-RP2, BSY1040-SS12-LP3, BSY1040-S14-RP6, BSY1040-S16-LP6, TCH1055-S1-RP3, TCH1055-S1-RP4, TCH1055-S2-LP2, TCH1055-S5-RP2	
Laetoli G1
(Fig. 2)	11	G1-23, G1-25, G1-26, G1-27, G1-31, G1-33, G1-34, G1-35, G1-36, G1-37, G1-39	
Laetoli tracks (Fig. 3)	22	G1-23, G1-25, G1-26, G1-27, G1-33, G1-34, G1-35(M), G1-36, G1-37, G1-39, G2-18, G2-26, G2-27, G2-28, G2-29, L8-S1-1, L8S1-2, L8S1-3, L8S1-4, TP2S1-1, TP2S1-2, TP2S1-4	
J. curtedulensis
(Fig. 4B)	4	SCR1500-T1-L8 (h), SCR1500-T1-L7, SCR1500-T1-R7, BSY1050-TR2-R3	
J. curtedulensis
(Fig. 4C)	7	SCR1500-T1-R3, SCR1500-T1-L4, SCR1500-T1-R4, SCR1500-T1-L5, SCR1500-T1-L7 (p), SCR1500-T1-R7 (p), SCR1500-T1-L8 (h)	
M. transjuranicus
(Fig. 5)	7	TCH1030-T6-L1 (h), TCH1030-T7-L2, BSY1040-T1-R1, TCH1025-T2-L1, TCH1030-T2-R2, TCH1030-T2-L3, BSY1035-T6-L2	

Figure 1 Example of a stat-track-based morphological comparison based on tiny and small sauropod tracks from the Ajoie ichnocoenosis.

(A) Mean stat-track of tiny sauropod tracks based on 10 specimens with a mean pes length of 11.6 cm. Dark blue indicates the highest part, dark red the deepest part of the tracks. Scale bar: 10 cm. (B) Mean stat-track of small sauropod tracks based on nine specimens with a mean pes length of 36.9 cm. Dark blue indicates the highest part, dark red the deepest part of the tracks. Scale bar: 10 cm. (C) Superimposition of mean stat-tracks contours of the tiny (black) and small (red). (D) Maximum difference stat-track between the mean stat-tracks. The colours quantify the deviation between the models along the z-axis (values in mm). The highest differences are concentrated in the depth and position of digits I and II. (E) Superimposition of contours of the tiny (black) and small (red) best tracks. (F) Maximum difference stat-track between the best tracks. The darker the colour, the larger the difference. The colours quantify the deviation between the models along the z-axis (values in mm). The biggest difference is located on the position and depth of digit I and digit II claw marks, and on the depth of digit I. “I” and “II” indicate digit I and digit II, respectively.

Late Jurassic sauropod tracks (Ajoie ichnocoenosis, Switzerland)

With this case study, we show how stat-tracks can be used to compare tracks from the same ichnocoenosis characterized by similar morphologies but (very) different sizes in order to identify if there were affinities among the tracks and if it was possible to determine a single or multiple trackmakers independently from the dimensions of the tracks.

Sauropod tracks are very common in the Late Jurassic Ajoie ichnocoenosis (Canton Jura, NW Switzerland) and vary in size from tiny (pes length < 25 cm) to large (PL >  75 cm) (Marty, 2008; Marty et al., 2010; Marty et al., 2017; Belvedere et al., 2016). The tiny (PL < 25 cm, the smallest around 10 cm in mean pes length) and small (25 < PL < 50 cm) tracks present very strong similarities in pes digit configuration and position, as well as in overall pes (and manus) morphology, and were therefore chosen for this analysis. Only the best-preserved (grade >2 on the scale of Belvedere & Farlow, 2016) pes tracks available in the PALA16 collection (see Supplemental Information 1) were digitized and used. This decision was taken to maximise the quality of the comparison from a morphological point of view, for which purpose the highest morphological detail possible was needed. A total of ten tiny and nine small pes tracks were used to generate the mean stat-tracks for each size class (Figs. 1A, 1B). Accordingly, in the following comparison, variations due to different individual trackmakers, to different locomotion styles, and due to differences in substrate properties are included. The registration contours superimposition (Fig. 1C) and maximum stat-tracks (Fig. 1D) both show pronounced similarities amongst the tracks. The differences are isolated only to the depth, especially of digit I and II (the darker areas in the Fig. 1D). The resemblance is so similar that, without taking into account the considerable size difference (the largest is around four times longer than the smallest), all these tracks could fall in the range of intraspecific or even intra-trackway variation (Razzolini et al., 2014; Razzolini et al., 2017; Lallensack, Van Heteren & Wings, 2016), despite the fact they are the result of the comparison of different trackways (different animals, trackmakers) and from different ichnoassemblages and track levels. To highlight also the slightest differences, which might be mitigated by using the mean stat-tracks based on a relatively large sample size, the best-preserved tracks for each size class (BSY008-S25-RP6 for the tiny and BSY008-S12-RP2 for the small tracks) were also compared (Figs. 1E, 1F), showing an even greater similarity in the general morphology, with the only substantial difference related to the depth, especially in the heel and digit I areas (darker areas in Fig. 1F).

To conclude, the studied sauropod pes tracks from the Ajoie ichnocoenosis are extremely similar and consistent in shape, despite the different weight (due to the difference in size) of their trackmakers, different animals having left the tracks, and despite (slight) differences in substrate properties (different track levels). The generation of stat-tracks highlights differences in impression depth of the digits (as expected given the pronounced difference in weight and the similar substrate), the position and orientation of the claw marks, which result in the mean stat-tracks in a slightly obliterated and vague claw mark for both the small and tiny tracks. Another difference is the size of the depression behind digit I, in the proximal part of the track, which is slightly bigger and elongated in the small tracks when compared to the tiny ones (Fig. 1D). The same difference in the proximal (heel) area is present in the best-track comparison, whereas the difference in the position of digits and claws is minimal. This implies that the substrate had a very similar rheological response to the different weights of the animals to preserve the same degree of details in two different track size ranges. Also, it is concluded that the two size classes clearly belong to the same ichnogenus and ichnospecies (yet to be defined). Moreover, stat-tracks can better support the presence of different size (age) classes of a single trackmaker species than the simple description of the presence of ‘similar’ morphologies in different size classes (Belvedere et al., 2016).

Pliocene hominin tracks (Laetoli, Tanzania)

Stat-tracks have recently been used to examine the Laetoli tracks and make comparisons with the tracks made by other hominin species (Crompton et al., 2012; Bennett et al., 2016a; Bennett et al., 2016b). In this case study, we extend the use of mean stat-tracks by including the additional data recently published from sites adjacent to the original trackways (Masao et al., 2016), highlighting also the importance and opportunity of using digital models produced and shared by other researchers. The inclusion of this new data adds significantly to the mean forms produced to date.

Laetoli is probably the most iconic of all human trackways and was first discovered and excavated in the late 1970s and now dated to 3.66 Ma (Deino, 2011). For many people they provide one of the earliest direct sources of evidence for hominin bipedalism (Leakey & Hay, 1979; Leakey, 1981; Leakey & Harris, 1987), although the degree to which the biomechanics of the trackmaker resemble those of modern humans has been subject to extensive debate and remains controversial (e.g., Bennett et al., 2016a; Hatala et al., 2016). The year 2016 saw the number of individual trackmakers at Laetoli rise from three, with only one useable trackway, to a total of in excess of five trackways. The discovery of additional tracks published in 2016 (Masao et al., 2016) has further increased the value of this already important site, particularly since the new tracks yield a more varied insight into the potential height and weight of the trackmakers assumed by most researchers to be Australopithecus afarensis (White & Suwa, 1987; Harcourt-Smith, 2005).

Figure 2 A series of eight contour maps for means stat-tracks of the G1-Trackway generated by different operators.

(A–K) Seven operators were asked independently to create a mean track using DigTrace from 11 individual tracks. (L) The variation between these mean tracks is small and operator variance can be removed completely by creating a ‘super’ mean combining each individual mean stat-track. Contour interval is 1 mm. See Table 1 for more information of the specimens used.

Figure 3 Mean stat-tracks for hominin tracks at Laetoli.

The trackmaker is generally accepted to have been Australopithecus afarensis. (A) G1 trackway. (B) G3 trackway. (C) L8 trackway. (D) M9 trackway. (E) TP2 trackway. (A–E) are all illustrated at the same scale. (F) Mean stat-track of Laetoli tracks, which could be used as mediotype in a future revision of Praehominipes laetoliensis. (G) Comparison between the Laetoli mediotype (red) and a modern human track (black). Note that the comparison is scale-free. See Table 1 for more information of the specimens.

Formal ichnotaxonomy has not been widely applied to hominin tracks; however, the Laetoli tracks were used for this purpose by Meldrum et al. (2011) (Praehominipes laetoliensis), following the lead set by Kim et al. (2008) who used the tracks preserved in volcanic mud (Lockley et al., 2007; Schmincke et al., 2009), and under cover in a museum at Acahualinca (Nicaragua) to define the ichnotaxa Hominipes modernus for modern human tracks. The value of these ichnotaxa per se is perhaps questionable, although it is correct to question why hominin tracks should be an ichnological exception. Human track morphology varies with age, gender, and body mass. The latter is a function of nutrition and environmental/climatic conditions and body size ratios (i.e., foot length to height) are known to vary also with ethnicity/race. They may also vary between different hominin species with both biomechanical and anatomical variations possible (Bennett & Morse, 2014). Mean tracks for the G1-trackway at Laetoli were first calculated by Crompton et al. (2012) using pSPM and subsequently by Bennett et al. (2016a) and Bennett et al. (2016b) using DigTrace. One potential area of subjectivity is the placement of the landmarks used for registration. Figure 2 addresses this issue; each of the mean tracks presented (Figs. 2A–2K) was generated by an independent operator and the similarity between each of these mean tracks is clear. It would also be possible to generate a mean track on the basis of these means thereby reducing operator variance further (Fig. 2L). In addition, Bennett et al. (2016b) used DigTrace to separate the composite tracks of the G2-trackway which is composed of superimposed tracks made by at least two (maybe three) trackmakers walking in line. Using a combination of individual tracks from the G1, G3, L8, M9, and TP2 trackways, and published by different authors (Bennett et al., 2016a; Masao et al., 2016), it is possible to establish a mean stat-track for each trackway (Figs. 3A–3E) to supplement the formal ichnotaxa proposed by Meldrum et al. (2011) accounting different sizes and morphologies of the same hominin ichnoassociation. Figure 3F shows a mean mediotype for the G1-Trackway at Laetoli based on the 11 topologically most complete tracks. In this case, the value in a mediotype lies in providing a scientifically agreed mean or representative track with which both intra- and crucially inter-site comparisons can be made (Bennett et al., 2016b). A revision of P. laetoliensis should therefore include the recent discoveries and, if worth, add new type specimens and create a mediotype for the ichnotaxon. Using the mean stat-track generated from the various tracksites, allowed a meaningful comparison with a modern human footprint (Fig. 3G). This provides continued perspective on the degree of medial transfer in the latter stages of stance within the Australopithecines (Bennett et al., 2016a; Hatala et al., 2016).

Late Jurassic theropod tracks (Ajoie ichnocoenosis, Switzerland)

These examples serve to underline the ichnotaxonomical potential of mediotypes and stat-tracks. In the focus are two new, recently described, ichnotaxa: Jurabrontes curtedulensis. (Marty et al., 2017) and Megalosauripus transjuranicus (Razzolini et al., 2017), both erected on Late Jurassic tracks from the Ajoie ichnocoenosis (Canton Jura, NW Switzerland).

Jurabrontes curtedulensis

This is a giant (PL > 50 cm) new theropod ichnogenus and ichnospecies, based on very well-preserved material (grade 2.5 to 3 of Belvedere & Farlow, 2016). Four type-specimens (holotype, three paratypes) were used to define this ichnotaxon (Fig. 4A), and for the first time the publication was accompanied by a mediotype generated through DigTrace. Different approaches were considered to generate the mean tracks: the mediotype (Fig. 4B) was based on the four type specimens, which differ especially regarding their impression depth due to differences in substrate properties and thickness. In the mediotype, all key features (including the very faint impression of digit III’s proximal phalangeal pad) of the description of the ichnotaxon are visible, confirming the morphological observations and descriptions made with standard methods (direct observation of the specimen, outline drawings, depth maps). A second set of mean and median stat-tracks (Fig. 4C) was generated from the holotype trackway to determine intra-trackway variability. The holotype trackway is composed of 11 clear and continuous tracks, and the mediotype was calculated using the seven digitized tracks (either gathered in 2011 with a laserscanner or in 2016 through photogrammetry).

Figure 4 Jurabrontes curtedulensis holotype and mediotypes.

(A) Photograph of the holotype (SCR1500-T1-L8). Scale bar 20 cm. (B) Mediotype generated from the 4 type specimens. (C) Mean stat-track generated from 7 tracks of the trackway including also the holotype and two of the paratypes. See Table 1 for more information of the specimens used. “II”, “III” and “IV” indicate digit II, digit III and digit IV, respectively.

The derived stat-tracks show a very conservative shape and highlights all the key features of the new ichnotaxon, despite the variation in depth and preservation of the individual tracks and even by incorporating tracks that were not good enough to be considered as type-specimens.

Megalosauripus transjuranicus

This new ichnospecies was erected based on large theropod tracks that were frequently found on tracksites and levels of the Ajoie ichnocoenosis. This ichnotaxon presents some peculiar characteristics (e.g., the large proximal pad of digit IV) that identify it as a new ichnospecies (Fig. 5A) of the ichnogenus Megalosauripus (Razzolini et al., 2017).

Figure 5 Megalosauripus transjuranicus photo, mediotype and stat-track.

(A) Photograph of the holotype (TCH1030-T6-L1) of the new ichnospecies M. transjuranicus. Scale bar 20 cm. (B) Mediotype generated from the 7 type specimens of M. transjuranicus. (C) Standard deviation stat-track of all 7 type specimens. The colours quantify the deviation among the models along the z-axis (values in mm). “II”, “III” and “IV” indicate digit II, digit III and digit IV, respectively. See Table 1 for more information of the specimens used.

Even though the seven type specimens are from different tracksites and track levels (i.e., they are not coeval but may have been left within some hundred to ten thousand years of difference), and the fact that also the preservation varies (one of the paratypes is preserved as a natural cast), the mediotype (Fig. 5B) exhibits the key features of the ichnospecies. The standard deviation stat-track (Fig. 5C) supports these similarities, as most the of the differences among the type specimens is located in cracks, which are not present in all samples, and areas that are affected by a high degree of mobility of the tridactyl foot during locomotion (e.g., digit III distal part, or digit IV width) (Belvedere, 2008; Castanera et al., 2013; Lallensack, Van Heteren & Wings, 2016).

An important application of stat-tracks is to validate ichnotaxonomical assignations or relationships by comparing stat-tracks and mediotypes of different specimens and ichnotaxa. The ichnogenus Megalosauripus has often been used as wastebasket in ichnotaxonomy, and it includes tracks that are quite different one from another. In addition, the ichnogenus has issues about its validity (see Lockley, Meyer & Santos, 2000; Thulborn, 2001 for the Megalosauripus-Megalosauropus dispute). To show the potential of stat-track- and mediotype-based analyses, a comparison is shown here (Fig. 6) between the M. transjuranicus mean mediotype (Fig. 5B) and two other tracks, one belonging to the known Megalosauripus ichnospecies M. teutonicus (Fig. 6A), and the other to a Megalosauripus isp. track (Fig. 6C) from the Late Jurassic of Morocco. In the first comparison, the standard deviation stat-track (Fig. 6B) shows only few similarities; it shows a lower deviation (lighter colour) around digit IV and digit II, whereas digit III presents higher differences, and it’s almost impossible to identify. Digit IV seems to maintain the shape of the M. transjuranicus mediotype: this is probably due to a combination of similarities between the two samples, but highly biased by the completely different preservation of the two specimens. Considering all these aspects, the comparison through stat-track supports and validates the interpretation of Razzolini et al. (2017) in the attribution of material from the Ajoie ichnocoenosis to a new ichnospecies different from M. teutonicus (Keaver & De Lapparent, 1974). Despite the fact that they are assigned to the same ichnogenus and that the best available specimens were used, the comparison clearly indicates that these two tracks do not have so much in common. This underlines how important it is that ichnotaxa are erected only on particularly well-preserved tracks, and that a detailed revision of the known ichnotaxa is needed. The second comparison (Fig. 6D) emphasizes the great resemblance between tracks from the Ajoie ichnocoenosis and from Morocco. The standard deviation stat-track shows marked differences (very dark) in the position of the digit III claw impression (the dark triangle ahead of the digit), which is isolated and not connected to digit III in the Moroccan track, and in the depth of the inner part of digit III. This can be explained by the different depth of the two digits, and by the much steeper walls of the Moroccan specimen. Another difference occurs in the position and shape of digit II, that can be explained with the fact that in the Moroccan tracks digit II shows some dragging marks. Some differences occur also in digit IV, where the less distal phalangeal pads seem different from M. transjuranicus. As for digit III this can be explained with the different depth and width of digit IV of the 2 tracks (Moroccan track is deeper, the Swiss is wider) and with the steeper track walls of the Moroccan specimen. Despite these differences, which is worth noticing are limited to few millimetres (Fig. 6D), there is a high similitude between the tracks suggesting that not only the interpretation of the track as Megalosauripus isp. (Belvedere, 2008; Belvedere, Mietto & Ishigaki, 2010) was correct, but that, pending further analyses on a larger track sample, the Moroccan tracks could be addressed at least as M. cf. transjuranicus. This use of stat-tracks and mediotypes has a great development potential not only for ichnotaxonomical comparisons and attribution, but also to more accurately study the ichnotaxa present in different geographical areas or to better determine the time range and geographical distribution of a given ichnotaxon.

Figure 6 Examples of taxonomical applications of mediotypes and stat-tracks.

(A) Texturized three-dimensional mesh of a Megalosauripus teutonicus track from the Barkhausen tracksite, Germany. (B) Standard deviation stat-track between M. teutonicus and M. transjuranicus mediotype. The colours quantify the deviation between the models along the z-axis (values in mm). (C) Photograph of a megalosaurid track from Morocco (Deio CXXVIII/16 in Belvedere, 2008 and Belvedere, Mietto & Ishigaki, 2010). (D) Standard deviation stat-track between the Moroccan track and M. transjuranicus mediotype. The colours quantify the deviation between the models along the z-axis (values in mm). “II”, “III” and “IV” indicate digit II, digit III and digit IV, respectively. See Table 1 for more information of the specimens used.

Discussion

The case studies illustrate the potential of stat-tracks and mediotypes to help explore variability in track morphology. Essentially they help to: (1) define in a new three-dimensional way the ‘average’ or ‘typical’ morphology within a given sample of tracks, extremely important for ichnotaxonomical studies; (2) quantify the distribution of morphological variability within a population around the mean/median and identify which morphological/anatomical areas of a track are responsible for that variability; (3) compare both qualitatively and statistically ‘typical’ morphological distributions from different sites, ages, and/or trackmakers; (4) define individual, potentially important, departures (i.e., specific cases) from a typical morphology; and (5) explore how morphological variability changes with changes in biomechanics or substrate. Whatever methodology used to co-register tracks, the potential to enhance ichnological analysis is clear and we argue that this procedure should become a standard part of ichnological research to complement, rather than substitute, the traditional and well-established ichnological analyses.

That is not to say that there aren’t some challenges here. There is an ever-present risk that important but under represented features in a track are ‘washed-out’ by the averaging process and there will always be a role for considering, and perhaps emphasising individual tracks in making an interpretation. Equally there is a risk of ‘forcing’ different tracks into a single population. In the case of DigTrace careful consideration of the standard error, based on the fit of all the placed landmarks, is essential and statistical tools to map 95% confidence measures are in development. In fact, pSPM explicitly allows statistical assessments of this sort (Crompton et al., 2012). We would argue therefore that the benefits outweigh the potential risks of ‘averaging’ affects, which can also be mitigated by a careful selection of the specimens to investigate.

The study of similar tracks from the same locality or a selected stratigraphic range is reinforced by the use of stat-tracks that allows reliable and repeatable comparisons among different specimens. As such, population studies have the possibility to move towards a more accurate identification of the relationship between different trackmakers, as presented in Case Study 3.1 and 3.2. The possibility to gather the characteristic features of several different specimens in one and the same mean or median stat-track (or mediotype if referred to type material) allows mitigating, when not eliminating, extramorphological features from the description of the typical characteristics of a given track sample, thus providing a more accurate morphotype. The three-dimensional comparison of different morphotypes using their mean stat-tracks will make analyses of ichnoassemblages and ichnocoenoses more objective and trust worthier as they will be less affected by a researcher’s subjective interpretation of the tracks.

The study of the Laetoli tracks (Case Study 3.2) highlights the quality and the reliability of the landmark placing, which generates higher quality morphometrical analyses through a geometric morphometric application. Moreover, the possibility of working with such a reliable mediotype for the hominin tracks, allows a formal basis for interspecies comparisons at least at the genus level, for example comparing tracks made by Homo with those of Australopithecines, and, not last, for more detailed evolutionary studies.

Whether mediotypes have a role in formal ichnotaxonomy needs to be considered. Ichnotaxonomy is not without its philosophical and methodological problems because of the morphological variability often present within tracks and the potential for multiple track morphologies to be associated with a single trackmaker. In fact, it might be fair to say the ichnology community is polarised between those that favour formal classification of tracks and those that don’t and who emphasis the biomechanical and behavioural importance of tracks over their formal identification or trackmaker identification (e.g., Gatesy & Falkingham, 2017). It is worth however exploring the potential significance of mediotypes in terms of ichnotaxonomy.

The Linnaean classification of fossil vertebrate tracks (footprints) is an agreed, but often uncertain practice (Demathieu & Demathieu, 2003). Given the peculiar nature of tracks as the result of the combined interaction of foot morphology, substrate properties, locomotion and behaviour, unlike conventional palaeontology finding a ‘morphologically-perfect’ specimen is quite rare. Morphological and taxonomical studies should be carried out therefore on the largest number of (morphologically well-preserved) specimens available (Sarjeant, 1989), in order to better understand the influence of extramorphological factors on the definition and description of an ichnotaxon. In the best cases ichnotaxonomical descriptions are accompanied by photographs and illustrations of the type-specimens. These however, do not always represent the sum of the characteristics of an ichnotaxon, but simply the features of each type specimen included. We argue that the use of 3D data may assist in correcting this omission.

The usual way of giving names to tracks consists of a binomial combination of the (ichno)genus and (ichno)species names (Bertling et al., 2006). As for other zoological disciplines, the nomenclature process follows rules established by the International Commission of Zoological Nomenclature (ICZN) published in its code (ICZN, 1999). The ICZN code defines the holotype as “the single specimen upon which a new nominal species-group taxon is based in the original publication” (ICZN, 1999, Art. 73) and paratypes as “any remaining specimens of the type series” (ICZN, 1999, Art 73, Recommendation 73D).

Due to this complexity, vertebrate ichnogenera and ichnospecies should be defined on morphological criteria of the track (Thulborn, 1990) rather than on the supposed systematic affinity of the trackmaker, and they may change if the animal’s behaviour changes (Sarjeant, 1990). Moreover, it is good practice to capture all potential ichnotaxa formed by a single trackmaker. For these reasons the ICZN code, since 1979, has exempted ichnotaxa from zoological taxonomy, stating that an ichnotaxon does not compete in priority with a taxon established for an animal, even if the animal may have formed the track (ICZN, 1999, Art. 23.7).

The erection of a new ichnotaxon should also consider the ‘Ten palaeoichnological commandments’ of Sarjeant (1989). These commandments stress above all: the importance of basing a new taxon on trackways (i.e., a track population) and not on isolated tracks (I); support for a new ichnotaxon should be based on detailed illustrations, photos, and digital models (IV, V); and provide unambiguous diagnoses of the trackmaker where possible (VIII). It follows that an ichnotaxon should present the main morphological characteristics and the key features that distinguish it from any other. This should be based on creating, from different type specimens (holotypes and paratypes), an average description including the key features, which cannot normally be supported by a single illustration or photograph as it is an abstraction based on multiple tracks. Therefore, the diagnosis of a new (ichno)taxon is the description of the key features present in all type specimens, with remarks on the most characteristic ones. Illustrations (outline drawings, photographs, 3D models) on the other hand, only illustrate distinct specimens, which may not include all of the described typical features. In a certain sense, while the description makes an average description of the specimen, illustrations of single specimens only represent some peculiar aspects. Despite being formally right, in a complex and very qualitative discipline as vertebrate ichnology, the attribution of a track to a certain ichnotaxa is often based on (highly subjective) morphological and graphical comparisons. Moreover, there might be differences amongst the holotype and the paratypes related to differences in substrate properties and other extramorphological factors that can make the comparison with other specimens even more difficult. Olsen & Baird (1986) tented a solution for this issue, by creating ‘composite tracks’ putting together the most important features of various key specimens of a single ichnotaxon (e.g., Atreipus), but the output was still a bi-dimensional outline drawing.

Over the last years, the use of three-dimensional models has spread widely in ichnology and is already becoming a standard for sharing data, also thanks to the diffusion on cloud services (e.g., Figshare). Nonetheless, so far only a few publications have provided extensive digital data at the moment of the institution of a new ichnotaxon (e.g., Razzolini et al., 2017; Marty et al., 2017). However, despite carrying more information than bi-dimensional images, three-dimensional models (as photographs and outline drawings) only represent a specific specimen, which may not contain all of the key features of the ichnotaxon. The use of mediotypes is a powerful tool for ichnologists to summarize and illustrate all key features of an ichnotaxon and to statistically support the observations made on the actual specimens and the descriptions of an ichnotaxon. The development of such tools like DigTrace and other similar examples, the diffusion of the concept and use of mediotypes together with the increased application of three-dimensional and quantitative methodologies in vertebrate ichnology can, in the near future, allow a new quantitative and statistically-based approach in ichnotaxonomy.

It might also soon be possible to establish variation thresholds between studied specimen(s) and the reference ichnotaxon (holotype, paratypes, mediotypes). Therefore, if the specimen’s values are within the thresholds it can be assigned to the reference ichnotaxon with a higher confidence, whereas when the values don’t pass the threshold, the specimen shouldn’t be referred to that ichnotaxon. Given the complexity of track identification and interpretation, a purely quantitative and statistical attribution is not reliable, as it does not consider those very punctual differences caused by for example taphonomy (Marty, Strasser & Meyer, 2009) or excavation (partially present track fills) and weathering damages, which can be observed only in the actual specimen (in some cases not even in the digital replicas). Therefore, these quantitative tools and analyses should always be accompanied by a classical descriptive and qualitative approach, that will consider those features that do not affect the ichnotaxonomical interpretation, and, eventually serve to identify some of the extramorphologies as observed on 3D models. We suggest that mediotypes and the other stat-tracks have the potential to have a deep impact on ichnotaxonomy, although, as ‘plastotype’, ‘digitype’ has not yet been formalized in the ICZN, and can at the moment not be solely used to erect a new taxon.

Conclusions

The concept of the stat-track (and mediotype) formalises a current trend in vertebrate ichnology, namely the description and comparison of morphological variability via 3D track data. It is based on the idea of the co-registration of different tracks to create mean/median representations of a track population, to allow comparisons of those mean/median tracks between sites, substrates and trackmakers and crucially to allow departures from mean/median tracks to be examined. Stat-tracks also permit the comparison though different statistical approaches, of distinct tracks and the generation of virtual models of the differences among specimens, which can be used to quantify and, also, visualize the similarities among different specimens. There are a number of tools, including DigTrace used here, available to aid this type of analysis and we encourage the development of further tools. We argue that mediotypes increase the value of ichnotaxonomical interpretations. Consequently, mediotypes are produced from type specimens only and represent the best three-dimensional approximation of the diagnosis of a given ichnotaxon. Whether the mediotype concept is validated by the ICZN, or not, like the ‘digitype’ concept, in the future we believe it will have a deep impact on ichnotaxonomy and should become a standard for the description of new and validation of existing ichnotaxa. Finally, stat-tracks have an important role in understanding and crucially documenting the morphological variability of tracks produced by a single trackmaker under varying conditions and circumstances, and this will enhance the understanding of the locomotive and behavioural range of a given ichnospecies.

Supplemental Information

Supplemental Information 1 Supplemental Information on methods and geological settings

The supplemental file includes information about the methodologies used in this paper to collect and analyse three-dimensional data and more information on the geological context of the studied areas, i.e., the Laetoli tracksite (Tanzania) and the Ajoie ichnocoenosis (Switzerland).

Click here for additional data file.

Supplemental Information 2 Raw data repositories

The document includes the link to all the repositories where the raw data used in this work are located.

Click here for additional data file.

We thank Novella L. Razzolini, Christian A. Meyer, Diego Castanera, and Michela Contessi for discussions on the idea of the mediotype concept and about its possible use in ichnotaxonomy. MBv and DM thank all technicians, photographers, geometers, drawers, collection managers, and preparators of the PALA16 that were involved during the excavation and documentation of the tracksites and during the set-up and organization of the track collection, as well as the scientific staff of the PALA16 and JURASSICA Muséum for various stimulating discussions and valuable input. The National Museum of Kenya is acknowledged with thanks for providing access to the first generation of cast of the Laetoli tracks in 2008. MRB would also like to thank Robin Crompton, Karl Bates and Todd Pataky for useful discussion about whole foot methods. A copy of DigTrace can be obtained from http://www.digtrace.co.uk. Finally, we thank the editor Jérémy Anquetin, and the journal reviewers Anthony Romilio and Sebastian Voigt or their insightful feedback and comments that considerably improved the quality of this manuscript, and the four anonymous reviewers of the previous submission. In addition, we wish to thank all the participants at the ICCI 2017 meeting which, together with the reviewers, helped in developing the mediotype and stat-tracks concepts.

Additional Information and Declarations

Competing Interests

Author Contributions

Data Availability

The authors declare there are no competing interests.

Matteo Belvedere and Matthew R. Bennett conceived and designed the experiments, performed the experiments, analyzed the data, contributed reagents/materials/analysis tools, wrote the paper, prepared figures and/or tables, reviewed drafts of the paper.

Matthew R. Bennett conceived and designed the experiments, performed the experiments, analyzed the data, contributed reagents/materials/analysis tools, wrote the paper, prepared figures and/or tables, reviewed drafts of the paper.

Daniel Marty conceived and designed the experiments, analyzed the data, contributed reagents/materials/analysis tools, wrote the paper, reviewed drafts of the paper.

Marcin Budka conceived and designed the experiments, performed the experiments, contributed reagents/materials/analysis tools, wrote the paper, prepared figures and/or tables, reviewed drafts of the paper.

Sally C. Reynolds analyzed the data, contributed reagents/materials/analysis tools, wrote the paper, reviewed drafts of the paper.

Rashid Bakirov performed the experiments, contributed reagents/materials/analysis tools, reviewed drafts of the paper.

The following information was supplied regarding data availability:

Belvedere, Matteo; Bennett, Matthew R; Marty, Daniel; Reynolds, Sally; Budka, Marcin; Bakirov, Rashid (2017): Supplementary material ‘Stat-tracks and mediotypes: powerful tools for modern ichnology based on 3D models’. figshare. https://doi.org/10.6084/m9.figshare.5241046.v1.

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
