# Peer review of "Stat-tracks and mediotypes: powerful tools for modern ichnology based on 3D models"

_PeerJ, doi:10.7717/peerj.4247_

## Round 0.1 · original submission · Major Revisions

I apologize for the unusual time your submission stayed in review. I waited for a third opinion, which is now more than three weeks overdue and will, despite several reminders, probably never come. The two reviewers agree that there is definitely something interesting in your study, but they both recommend major revisions (see their respective detailed reports).

One of the main issues to consider is that there is uncertainty in the way you propose to use the concept of mediotype. This is something that confused me also when reading your manuscript. It is as if there could be a ‘mediotype sensu stricto’ that could be an interesting addition to the definition of new ichnotaxa (but see all the potential problems listed by Anthony Romilio), and a ‘practical mediotype’, which is basically a tool to produce mean/median shapes to be used for comparisons (and there also there are potential problems relating to the choice of the tracks, etc.). In my opinion, this is really something that should be clarified before publication. I would suggest that the term ‘mediotype’ should be restricted to the definition of new ichnotaxa and you should clearly define what it is and how it is produced (Anthony Romilio is right about the potential confusion the term ‘type’ might bring, so maybe a different term would be better). The fact of producing a mean/median shape for comparative purpose (not just between type series) should be called differently to avoid confusion, and described not as a concept but as a tool (and there you need to reply to the problem of the subjective choice of the tracks).

Therefore, I tend to agree with the reviewers that your manuscript needs some major reworking before it can be accepted for publication. I have some additional comments in the attached annotated manuscript. Notably, you will note that Table 1 is missing. I also think that some parts of the text (especially the Discussion) are sometimes difficult to follow, but maybe once the ‘mediotype’ definition is clarified these issues will be solved naturally.

·

Basic reporting

This paper introduces the term 'mediotype’, which incorporates a multitude of digital 3D data files relevant to the field of (fossil) track research. The word is enclosed by inverted commas in the title and is a practice that should be continued throughout the body of work. I have no further comments to add to their Basic Reporting as are very well structured and of publication quality.

Experimental design

The authors identify a significant gap in palaeoichnology, that is how to obtain track data that can be utilised in quantitative comparative studies (albeit in a series of paired analyses). They succeed in their approach by utilising the powerful, innovative, and newly developed DigTrace software. The authors provide a series of case studies that are relevant for the application of the method, by which the findings are very clearly documented. This approach is of very high significance for this research field (particularly dinosaur tracks, and is one that I am confident that progressive palaeoichnologist are/will be eager to employ.

As clearly stated by the authors, this is not a methods paper. The aim of the paper is to introduce ‘mediotype’ as a term for use by the wider palaeoichnological community to incorporate as range of ‘statistically-generated three-dimensional track models [SG3D]’. While ’mediotype’ is a pleasant term that is simply to use, I do not think the term will be received unfavourably by palaeoichnologists for the following reasons :
• Etymology implies a type specimen: ‘Mediotype’ adds to now a growing list of ICZN non-formalised terms (e.g., ‘digitype’, ‘plastotype’). Issues of repositories and accessibility to ‘type’ specimens will also come into play. Line 423 "unlike conventional palaeontology finding a 'morphologically-perfect' specimen is almost impossible" I am of the opinion that there will be strong resistance for the digital construction of an 'ideal' by any means. The authors "argue that the use of 3D data may assist in correcting this omission" which may be appropriate in for analyses but I disagree that type specimens should be a chimera of track data, nor would this be acceptable for type specimens for other research fields.

• Etymology implies ‘averaging’ of data: In practice, ‘average’ data sets (mean, median) appear to be the preferred documentation source for use in this paper (supplied figures, raw data). While I am in agreement that such track data are perhaps the most useful data sets for palaeoichnologists, the fact that their term encapsulates non-averaging statistical data makes use of the term non-inclusive and potentially misleading.

• Implies a single specimen: Even ‘digitype’ and ‘plastotype’ inferred specimens limited themselves to single representation. ‘Mediotype’ include six csv file types are generated with the DigTrace software (mean, median, maximum differences, minimum differences, standard deviation, point-to-point comparisons) most of which are not elaborated further after their mention in the paper.

• Implies that only type specimens can be used in the generation of SG3D track data whereby limiting application (e.g., just as ‘digitype’ is limited to type specimens), a restriction that is almost certainly not the intention of the authors.

Validity of the findings

The placement of landmarks can be problematic for tracks that are, by their very nature, highly variable abiotic structures. Although While the software the authors show preferences for using (Digtrace) displays significant robustness around this issue.

One aspects that is confusing it the greyscale visualisation and interpretations of the standard deviation ‘mediotypes’ shown in figures 1D, F; 5C; 6B, D. Here, white indicates low deviation and black indicates high deviation. Yet in each example that the authors state in-text that there is low deviation the plantar impressions are darker than the surrounding surface. In the example that the authors state in-text as high deviation (e.g., figure 6 B) most of the plantar impression is white and surrounding surface is darker. In these cases, the data contradicts the authors claim. This is particularly important to resolve as readers that become users of the Digtrace software may face the same issues.

Otherwise very well structured.

Additional comments

This is an important, well-written paper for dinosaur palaeoichnology and has the potential to be a very positive and broad influence for future research in this field.
The use of the term 'mediotype' is with significant problems that can be resolved, for the most part, by substituting the non-ambiguous and more accurate term 'statistically-generated three-dimensional track model' (SG 3D) of 'statistically-generated digital surface model' (SG DSM).

For further comments see attached annotated pdf.

·

Basic reporting

No comment.

Experimental design

No comment.

Validity of the findings

No comment.

Additional comments

Dear all,

I should emphasize that I am conservative regarding the use of 3D imaging in vertebrate ichnology. Therefore I would probably be categorized a ‚traditionalist‘ following the terminology of the authors. I do not refuse the application of 3D imaging in vertebrate ichnology, but I just do not see any use of hitherto suggested methods in order to solve the main problems of studying fossil tetrapod footprints.
The morphology of tetrapod tracks is the result of the superimposition of anatomy, gait and substrate. A majority of vertebrate ichnologists prefers an ichnotaxonomy highlights anatomically-based characters of the imprint morphology and trackway pattern. Therefore, the main challenge to these researchers is to distinguish anatomical input of tracks and trackways from extramorphological characters (= influence of gait and substrate conditions).

In which way a ‚mediotype‘ sensu Belvedere et al. may help anatomically-focused vertebrate ichnologists? As far as I understand the authors, the ‚mediotype‘ is mainly a tool to quantify the differences between two or more tracks. The result is a number which nothing helps to interpret the reason for the calculated difference(s). I definitely see that I could introduce tresholds such as a similarity index, e.g. two tracks should be similar by 90% in order to assign them to the same ichnospecies. However, I do not know how much of the similarity/dissimilarity is based on anatomy, gait, and substrate. Thus, where is the progress? I totally accept that the ‚mediotype‘ is one out of several tools for vertebrate ichnologists but I doubt that it has the potential to revolutionize this paleontological discipline.

What to do with this manuscript? It may be worth to describe the ‚mediotype‘ method and its application as examplified by the authors. But I feel that the importance of the paper is overstated. My suggestions: I would go along with the proposal to introduce tetrapod ichnotaxa at the base of long and well-preserved trackways, only. Thus, the introduction of ichnotaxa (diagnosis, description, measurements of track and trackway parameters) could be completed by data from 3D imaging in a way that the similarity of tracks within each specimen of the type series is calculated separately. This might result in useful values for other workers who want to quantify the similarity of their own specimens to the type material. By calculating discrete specimens the authors avoid the accusation of arbitrary selection as they base their models (‚mediotype‘) on type specimens exclusively (that are used by ‚traditionalists‘ as well). I do not see any use to extend the data base of the ‚mediotype‘ beyond type material (according to the requirements on quality and quantity of new ichnotaxa).

I think it would be a good paper to describe and illustrate the process of introduction a new ichnotaxa including a ‚mediotype‘ in most detail (documentation of the type material, traditional diagnosis, measurements, description, 3D model, ‚mediotype(s)‘ production). In addition, it would be helpful to apply the ‚mediotype‘ to similar specimens from another locality to examplify the tool. Certainly, my suggestions are not so far from the content of the author’s manuscript. However, I feel that the manuscript and its ideas would probably find more acceptance when the methods are demonstrated at a single example and in most detail. A good example would be Robledopus macdonaldi Voigt et al., 2013, small reptile tracks from the Early Permian oft he New Mexico as the ichnotaxon is based on a type series and there is ambiguous material from other localities of the same or similar age in New Mexcio.

Finally, I welcome the idea of the ‚mediotype‘ if it will be restricted to type material. The authors should think about my suggestions to focus on a discrete and fully elaborated example. At the present stage, I do not see that the manuscript will be of much use to the vertebrate ichnology community.

Sebastian Voigt
2017-08-17

---

## Round 0.2 · Minor Revisions

I have sent back the revised version of your manuscript to the previous two reviewers. You will see that both still tend to express some doubts as to whether your work will significantly improve the practice of paleoichnology. Something is sure however, as pointed out by the second reviewer, this debate will not be settled by the ongoing discussion between the authors and the two reviewers only. Which is why I think it is important that this paper is eventually published and its merits or pitfalls evaluated by the wider community.

In this revised version, you have carefully separated the concept of the mediotype sensu stricto and the more general stat-track idea, which clearly makes things easier. The first reviewer still has some objections to which I would like you to have a chance of responding before we can proceed further. Please, consider these new comments and submit a second revised version of your manuscript.

After reading this revised version, I have a few minor comments to add (see below).
- Line 155: "two published software solutions", DigTrace is one, but which is the second one?
- Line 166: "It is important" (add "is")
- Line 307: "austhralopithecines" not in italics
- Lines 372–373: "M. transjuranicus" in italics
- Line 376: replace "suggest" by "suggesting"
- Line 711: Place "Sarjeant WAS. 1989" on a new line
- Caption of Table 1: Mention that the abbreviations are explained in the Supplementary material (not sure if it is a good placement though)

·

Basic reporting

no comment

Experimental design

no comment

Validity of the findings

I am a supporter of the use digital methods to evaluate the 3D topography of track surfaces, and view such approaches as providing much needed quantifiable data to a field of science that is prone to much subjectivity.

One of the main aims pf this paper is to provide information of a recently developed tool for quantifiable comparative track analyses. I feel biased in my desire to have methods, like those employed by these authors, to be widely used by dinosaur ichnologists, and while the paper provides some new results from previous published examples, it is unclear as to its novelty. The authors explain the use and strengths of the approach very well, but this is not methods paper that would be of benefit to a wider audience and facilitate the adoption of this method by researchers. Shortcomings. The quantifiable aspects of this approach can be made much clearer in the examples the authors use. For example, the values of standard deviation should be stated within the text for the respective ‘stat-tracks’, as well as shown as a calculated range of standard deviation along the figured grey-scaled 'stat-track'. Inclusion in track figures of digit impression number are features currently absent and would facilitate ichnological assessments. Currently the statistical variants are comparative via grey-scale colouring of tracks yet lack numerical data.

My earlier appraisal of this work expressed confusion associated with author interpretation of grey-scaled 'stat-tracks’. While the authors have now elaborated their explanation further the point I wished to raise has been lost, that I hope to clarify here. In regard to the grey-scaled 'stat-tracks’, it is repeatedly stated that lighter-to-darker colours indicate less-to-more deviation (respectively), yet appear to not adhere to this criterion in interpreting the data. For example: in assessing sauropod tracks the are “pronounced similarities amongst the tracks. The resemblance is so similar that, …all these tracks could fall in the range of intraspecific or even intra-trackway variation”. Yet the ‘stat-track’ (Fig. 1D) shows very light colouration outside the track region and dark grey-black within the track region, the latter indicating areas very highly variants and can contradicts the statement within the text. But this is not in isolation, occurring in figures 5 and 6. In regard to the figure 6 theropodan tracks, “the standard deviation stat-track (Fig. 6B) shows only few similarities” which contradicts the very light colouring of the impressions of the digits II and IV and metatarsophangeal pad. The authors explanation of these similarities as “highly biased by the completely different preservation of the two specimens”. It is not made clear to the reader whom may wish to employ the tool outlined in the paper how to distinguish topographical similarities/differences as ichnologically significant/insignificant. A more comprehensive evaluation is wanting.

The second aim of this paper is to introduce new terms for potential future use in ichnotaxonomy. In considering neologisms for a very specific discipline of science, I question if it would be preferable to seek a more specific journal for the publication of this paper. This may be within either a palaeoichnological journal, or preferably a bulletin specific for nomenclature, where the validity of their proposal can be more definitively evaluated.

·

Basic reporting

No complaints.

Experimental design

No comment.

Validity of the findings

No comment.

Additional comments

Dear all,

the revised version of PJ_18800 addresses most of my former cautions with this manuscript. I especially welcome the separation of „stat-track“ and „mediotype“.

It is still my opinion that the importance of paleoichnological studies is first of all limited by the quality of studied tracks and traces and only subsequently affected by the applied techniques.

In the end, however, application in other studies will be the best way to decide about the utility of the author’s concept. Therefore I recommend this ms for publication.

Sebastian Voigt
2017-11-27

---

## Round 0.3 · accepted · Accept

Thank you for these last corrections. As noted in my last decision, I think it is time this contribution see the light of day and is discussed more widely within your community. I am therefore accepting it for publication in PeerJ.

If I can make one last suggestion, I would strongly recommend that you opt to open the peer-review correspondence we have had on this manuscript. I think there is value for the community to see the comments that were made on your paper and the replies that you have provided.